Species turnover in plants does not predict turnover in flower-visiting insects

http://orcid.org/0000-0002-8073-2804 Simaika John P. 1 2 simaikaj@sun.ac.za
Samways Michael 3
Vrdoljak Sven M. 3
1 Department of Water Science and Engineering, IHE Delft , Delft, South Holland , The Netherlands
2 Department of Soil Science, University of Stellenbosch , Stellenbosch, Western Cape , South Africa
3 Department of Conservation Ecology and Entomology, University of Stellenbosch , Stellenbosch, Western Cape , South Africa
Giordani Paolo
Electronic publication date: 2018 Dec 21
Publication date: 2018
Volume: 6
Electronic Location ID: e6139
Received 2018 May 15; Accepted 2018 Nov 20
Copyright: © 2018 Simaika et al.
Copyright year: 2018
Copyright holder: Simaika et al.
License: This is an open access article distributed under the terms of the Creative Commons Attribution License, which permits unrestricted use, distribution, reproduction and adaptation in any medium and for any purpose provided that it is properly attributed. For attribution, the original author(s), title, publication source (PeerJ) and either DOI or URL of the article must be cited.
License URL: https://creativecommons.org/licenses/by/4.0/

Keywords: Anthophile, Conservation planning, Congruence, Insect conservation, Pollination, Surrogacy

Funding: National Research Foundation of South Africa (NRF) German Federal Ministry of Education and Research through the BIOTA Africa Project, the NRF, and Stellenbosch University European Union’s Seventh Framework Programme 606838 IHE Delft John P. Simaika and Michael J. Samways were supported by the National Research Foundation of South Africa (NRF). Sven M. Vrdoljak was supported by the German Federal Ministry of Education and Research through the BIOTA Africa Project, the NRF, and Stellenbosch University. This final version of this paper was written up while John P. Simaika received funding for research from the European Union’s Seventh Framework Programme (grant agreement no 606838), with additional support from IHE Delft. The funders had no role in study design, data collection and analysis, decision to publish, or preparation of the manuscript.

==============================
Congruence between plant and insect diversity is considered possibly useful in conservation planning, as the better known plants could be surrogates for the lesser known insects. There has been little quantification of congruence across space, especially in biodiversity rich areas. We compare here species richness, and turnover relationships between plants and flower-visiting insects across space (0.5–80 km) in natural areas of a biodiversity hotspot, the Greater Cape Floristic Region, South Africa. A total of 22,352 anthophile individuals in 198 species and 348 plant species were sampled. A comparison between the plants and anthophiles suggest significant concordance between the two assemblages. However, turnover was weaker in plants than in anthophiles. Plant turnover decreased with greater geographical distance between plot pairs. In contrast, insect turnover remained high with increasing geographical distance between plot pairs. These findings suggest that while patterns of plant diversity and distribution shape flower-visiting insect assemblages, they are not reliable surrogates. The conservation significance of these results is that specialist mutualisms are at greatest risk, and that set-asides on farms would help improve the functional connectivity leading to the maintenance of the full range of mutualisms.

Introduction

Current extinction rates are of great concern (Pimm et al., 2014), with the achieving of conservation goals requiring an understanding of threats, the location of these threats, and determining how to avert them (Joppa et al., 2016). Yet to make informed conservation decisions, it is rarely possible to measure and account for all components of biodiversity, and so conservation planning is often based on surrogates, under the assumption that unknown diversity should also be effectively conserved by strategies for more familiar taxa or habitats (Rodrigues & Brooks, 2007; Wiens et al., 2008; Grantham et al., 2010). Effective surrogacy relies strongly on the degree of congruence between the surrogate or indicator taxa and target groups.

Congruence between plant and insect diversity has been widely discussed, with varying results depending on region, taxa, and diversity measures used. Although plant diversity can predict herbivorous insect diversity (Novotny et al., 2006), the relationship may not necessarily hold for non-herbivorous insects (Fontúrbel, Jordano & Medel, 2015). Furthermore, tests of taxonomic surrogacy often produce contradictory results (Kremen, 1992; Prendergast, 1997; Duelli & Obrist, 1998; Van Jaarsveld et al., 1998; Osborn et al., 1999; Dauber et al., 2003).

Assessing the value of congruency for identifying surrogates is challenging (Lovell et al., 2007), as spatial scale (Favreau et al., 2006) and the history of focal groups such as plants and insects (Ponel et al., 2003), play important roles. While there may be congruence between focal groups at very large spatial scales (Lamoreux et al., 2006; McKnight et al., 2007), this may not be the case at the smaller scale of conservation planning at the local level (Ricketts, Daily & Ehrlich, 2002; Stork & Habel, 2014). Furthermore, for surrogacy to be effective, the focal groups must respond to environmental variables in a similar way and must be equally sensitive (Pharo, Beattie & Binns, 1999), and for conservation planning this includes responses to human impacts (Kirkman et al., 2012). In addition, there is the obvious factor that insects overall are more mobile than plants and can more respond quickly to environmental change, which has happened both in the deep (Ponel et al., 2003) and recent past (Hickling et al., 2006).

Flower-visiting insects are highly mobile and show a wide range of specificity to the plant species they visit, with interactions between plants and insects ranging along a continuum from highly specialised to generalised (Pauw, 2013). The needs of anthophiles are not only restricted to plants, and also include aspects such as resting, courtship and mating, ovipositioning or nesting, and avoiding death (Vanbergen et al., 2013; Goulson et al., 2015; Gill et al., 2016). This means that anthophiles are sensitive to a range of factors that may not affect plants.

The Greater Cape Floristic Region (GCFR) biodiversity hotspot has one of the highest levels of plant species richness and endemism in the world, with 69% of 9,000 recorded plants endemic to the area (Goldblatt & Manning, 2002). There is still disagreement on whether insect diversity matches the high levels of plant species richness in the GCFR. Giliomee (2003), after conducting a study on herbivorous insects in the GCFR, suggested that the GCFR is not proportionately rich in herbivorous insect species, the exception being a guild of endophagous insects. Giliomee (2003) attributed this to the sclerophyllous nature and chemical defences of the plants, considering them a poor source of food for insects. On the other hand, Wright & Samways (2000), in their study on endophagous insects on Proteaceae in South Africa, found proportionately high species richness of herbivorous insects in the GCFR. Similarly, Kemp & Ellis (2017) found similar numbers of herbivorous insects per Restionaceae plant species, but as the plant species is so high in the region, the number of herbivorous insect species is inevitably high. Furthermore, Procheş & Cowling (2006), comparing the diversity patterns of plant-inhabiting insects in the local fynbos vegetation to that of three neighbouring biomes, found that fynbos insects are diverse, and follow the generally established plant-insect herbivore diversity relationship, as suggested also by Hawkins & Porter (2003).

Conservation planning and priorities are well established in the GCFR, with much focus on spatial planning and goal-setting for conservation strategies (Cowling et al., 2003; Pressey, Cowling & Rouget, 2003). The basis of this work rests largely on broad habitat units (Cowling & Heijnis, 2001), which are defined using climate, topography, geology and vegetation, as well as some ecological processes. These methods are still limited to taxa such as plants for which the necessary, detailed distribution data are available. Yet we know comparatively little about the diversity and distribution of insects in any biodiversity hotspot, including the GCFR, nor how well they would be protected within these spatial planning frameworks.

It is essential for effective conservation planning to consider congruency across taxa, as the choice of organisms has a strong influence over the representativeness of protected area networks. Maximizing the representativeness of taxa is done by considering their changes of beta diversity in the landscape (Socolar et al., 2016). The beta diversity measure consists of two components, the nestedness and turnover of species communities (Baselga, 2010). Turnover across a landscape occurs when species are replaced, creating distinct assemblages by the addition of novel species. Taxa that exhibit high spatial turnover require patches of high quality habitat to help conserve their communities (Baselga, 2010).

Here, we compare species turnover of plant assemblages and closely associated assemblages of flower-visiting insects (anthophiles) in natural and semi-transformed habitats in the GCFR, across space from 0.5 to 80 km. We pose two key questions: (1) Are species richness and turnover of anthophiles comparable across space to that of plants in the GCFR; and (2) is the community composition of anthophiles in a distance group related to local plant community composition? This study therefore addresses two issues that have been previously overlooked: the interrelationship between two mutually dependent groups, and how this relationship is affected by geographic distance. This knowledge is instrumental for informing conservation planning in the region and elsewhere in terms of plants and pollinators.

Methods

Study sites

Permissions for conducting the study were obtained from the relevant authority at: Helderberg Municipal Nature Reserve and Hottentots Holland Provincial Nature Reserve—permission from Cape Nature (Permit No. 372/2003); Cordoba Wine Estate, Vergelegen Wine Estate, Diepklowe Private Nature Reserve, Elandsberg Private Nature Reserve—permission from manager/owner.

This study was conducted in the lowlands of the GCFR, which includes threatened habitats (Rouget, Richardson & Cowling, 2003). Six sites across part of the region, either in formally protected areas, or on farms where land had been set aside for conservation purposes, were selected (Fig. 1; Table 1). At each of these sites, between one and four area plots were selected, all below 400 m a.s.l., and which represented the heterogeneity of natural lowland habitats in the region. In total, 16 plots were used in analyses.

Figure 1 Study area map, experimental design and study sites.

Map of study sites within the Greater Cape Floristic Region (A). Black circles = Cordoba (CO1-2), squares = Elandskloofberge (EL1-4), triangles = Helderberg Municipal Nature Reserve (HE1-3), star = Klipfontein (KL1), crosses = Groenlandberg Conservancy (GB1-3), diamonds = Vergelegen (VG1-2). Experimental design at each plot (B). For each plot, three arrays of randomly arranged coloured pan traps (blue, red, orange, violet, white, and yellow) were placed in a configuration as shown. Elandskloofberge site (EL3) with natural vegetation and located within reserve (C). Elandskloofberge remnant site surrounded by canola (as seen in background) and wheat fields (D). Photography by Sven Vrdoljak.

Table 1 Descriptions and locations of study sites and plots used to assess complementarity of anthophile and plant species assemblages in the lowlands of the Greater Cape Floristic Region.

Site/plot	Description	Location	Status	
Elandskloofberge—Elandsberg Private Nature Reserve (EL), 4,000 ha in extent, remnants on neighbouring Bartholomeusklip farm. All sites within three km radius.	
EL1	Remnant adjacent to reserve. Surrounded by wheatfields	33.4482°S, 19.0272°E	Remnant	
EL2	Old field on border of reserve	33.4438°S, 19.0291°E	Transformed	
EL3	Natural vegetation within reserve. BIOTA observatory site	33.448°S, 19.0474°E	Reserve	
EL4	Remnant surrounded by canola and wheatfields. Some disturbance by feral pigs	33.4536°S, 19.0162°E	Remnant	
Helderberg Region—Helderberg Municipal Nature Reserve (HE), 396 ha in extent, and remnants on nearby wine estates Cordoba (CO) and Vergelegen (VG). All sites within a nine km radius.	
HE1	Firebreak on margin of reserve, adjacent to golf estate	34.059°S, 18.8772°E	Disturbed	
HE2	Natural vegetation in within reserve	34.0618°S, 18.8749°E	Reserve	
HE3	Natural vegetation on former plantation area	34.0573°S, 18.8676°E	Reserve	
CO1	Former vineyard, replanted with natural vegetation	34.0334°S, 18.8488°E	Transformed	
CO2	Fragment between current vineyards, moribund, with invasive grasses	34.0313°S, 18.856°E	Remnant	
VG1	60 ha patch of largely intact renosterveld, adjacent to vineyards.	34.0948°S, 18.8974°E	Remnant	
VG2	Area cleared of IAPs adjacent to vineyard. Recovering vegetation with invasive Echium plantagenium.	34.0886°S, 18.8935°E	Transformed	
VG3	Old firebreak, 40 m wide with natural vegetation between dense stands of Acacia mearnsii.	34.0763°S, 18.923°E	Disturbed	
Groenlandberg Conservancy—reserve site in section of the Hottentots Holland Provincial Nature Reserve at Klipfontein (KL), 42,000 ha in extent, and remnants on Diepklowe Private Nature Reserve and olive farm (GB). All sites within a 10 km radius.	
KL1	Large block of relatively undisturbed natural vegetation situated near Theewaterskloof dam	34.0546°S, 19.169°E	Reserve	
GB1	Relatively intact remnant adjacent to fallow wheat field.	34.1017°S, 19.2496°E	Remnant	
GB2	Firebreak in area of moribund, Elytropappus rhinocerotis	34.1035°S, 19.2462°E	Disturbed	
GB3	Disturbed, but recovering area of natural vegetation on ridge above farm.	34.1099°S, 19.2448°E	Transformed	
Note:

Table modified after Vrdoljak & Samways (2014).

Insect sampling

Sampling took place over a three-month period (September–November 2005) to coincide with the time of peak flowering at each of the 16 plots. Vegetation and flowering status were measured a day before transect sampling (see the next section on Vegetation transects for further details).

Insects were surveyed using coloured pan traps (Vrdoljak & Samways, 2012). Anthophiles in the GCFR (Picker & Midgley, 1996) and elsewhere (Campbell & Hanula, 2007; Saunders & Luck, 2013) show differential colour preferences to pan traps, so a range of colours were used: red, orange, yellow, blue, violet, and white. Polypropylene tubs (RL350; Marco Plastics, Alberton, South Africa), 115 mm diameter by 50 mm deep (350 ml volume), were painted with gloss enamel paint (Dulux SA, Alberton, South Africa). For each site, three arrays of six coloured pan traps were used, arranged in a cross-shaped configuration of three 50 m lines at each of the 16 plots, with the six colours arranged randomly at 10 m intervals on each line (Fig. 1).

Pans were elevated and set at the level of flowers in the surrounding vegetation, and half filled with water, with a little detergent added to reduce surface tension. Elevating pan traps to the level of the canopy where insects are actively foraging significantly increases catches (Tuell & Isaacs, 2009). Trapping was only on sunny days, from 08h00 to 17h00. Trapped insects were removed from the water and preserved in 80% ethanol for later identification. Initial identifications were to morphospecies (Oliver & Beattie, 1996), with scientific identification to species where possible using the entomology collection in the Iziko South African Museum, Cape Town. Here, we refer to both morphospecies and species as ‘species’. Appendix S2 lists all insect species identified in this study.

Vegetation transects

At each of the 16 plots, vegetation was surveyed the day before the first day of pan trapping. Vegetation composition, height and cover were measured over three, 50 m transects per plot. These vegetation transects were along the same line transects as the three pan trap lines at each plot. All plants that covered the transect line were measured (height in centimetre, length of transect in metre), identified to species level, and their flowering status recorded (not flowering, flowering, in bud, in seed). Open patches of ground were recorded and classified according to whether they were bare ground, rock, leaf litter or woody debris.

Statistical analyses

Species richness estimates

For insects, total abundance of each species per plot was calculated from the pooled data of six arrays per plot (three arrays × two sample days per plot). For plants, data from all three transects were pooled for each plot. This was done so as to calculate the total intercept distance covered by each plant species and ground cover category (i.e. the effective abundance in terms of area covered by each species or category) at each plot. Species richness was estimated using the EstimateS Version 8.0 software package (Colwell, 2009; Colwell et al., 2012), using the pan trap data (three arrays × two sample days per plot). Many different species richness estimators are available, each with their own combinations of precision and bias that affect their accuracy (Walther & Moore, 2005). Given that certain anthophiles were highly abundant in pan trap samples, an incidence-based estimator, the Incidence Coverage Estimator (ICE; Chao et al., 2000), was calculated for each plot, using 1,000 randomisations, with replacement. The same procedure for calculating the ICE was followed for the pooled (three transects) vegetation data for each plot.

Species diversity

The Shannon diversity index was calculated for both plant and insect data in PRIMER Version 6 (Primer-E Ltd, 2002; Clarke & Warwick, 2001). A covariance analysis was conducted to test the significance of the relationship between insect and plant diversity.

Pan and plot assemblage similarity

Plots were classified using the CLUSTER routine in PRIMER Version 6 (Primer-E Ltd, 2002; Clarke & Warwick, 2001). Cluster analysis was based on Bray–Curtis similarities of the square-root transformed vegetation and pan trap data for each plot, which grouped them according to similarity of their plant and anthophile assemblages, respectively. Plots were classed using a similarity profile (SIMPROF) analysis on the null hypothesis that a specific sub-cluster could be recreated by permuting the entry sites. Significant branches (SIMPROF, p < 0.05) were then used to class plots together. The results of the analyses are presented in Appendix S3.

Species turnover

Seriation is used to test for species turnover along a spatial gradient (Brower & Kyle, 1988; Clarke, Warwick & Brown, 1993), and thus is an effective tool for detecting trends in taxon turnover that may be present (Clarke & Warwick, 2001). The index of seriation is given by Rho (q), ranging from −1 to +1, and provides a p-value at the 5% significance level. Values closer to −1 or to +1 indicate low community similarity, while values close to 0 indicate high community similarity. The RELATE function in PRIMER V6 was used to analyse the non-random spatial serial correlation of each set of assemblage data (plants and insects) between all the elements of Sorensen similarity matrices. The Sorensen similarity matrices were calculated from Bray–Curtis similarity matrices calculated from presence–absence transformed species abundance data. The RELATE function was then used to calculate the Spearman rank correlation coefficient between the plant and insect assemblage datasets. The Spearman’s rank coefficient can range from 0 to 1, where 1 is a perfect match between sample relationships.

To test for spatial relationships and species turnover, Sorensen pairwise dissimilarity of insect and plant species was calculated in the package betapart (version 1.5.0; Baselga & Orme, 2012) using R (version 3.4.3). Non-linear regressions were then fitted to the plant or insect datasets. To obtain r- and p-values, data were linearized using log10.

Effect of plant species composition on insect composition

In order to test whether plant species composition has a significant effect on anthophile species composition, we used an redundancy analysis (RDA) approach developed by Kemp, Linder & Ellis (2017) in R (version 3.4.3), using the package vegan (version 2.4-6; Oksanen et al., 2018). Forward selection in RDA was used to assess the influence of Hellinger-transformed plant species abundance on insect composition. Only the plant species selected by the forward selection were retained. RDA was then performed on Hellinger-transformed insect species abundances, with eight plant species as constraining variables and geographical distance as the conditioning variable. Geographical distance was converted to a rectangular principal coordinate of neighbour matrix for this analysis. To test the significance of variables, a permutational ANOVA test was done on the RDA.

Vegetation structure and composition

To compare the effects of vegetation structure and composition of the plant assemblage on flower-visiting insect assemblages at each plot, a number of variables were compiled from the vegetation data (Table 2). Plant species composition at each plot was summarised using detrended correspondence analysis (DCA) in CANOCO Version 4.53 (Ter Braak & Šmilauer, 2004) as the detrended segment lengths reported by CANOCO (maximum segment length > 4) indicated that the data were unimodally distributed (Lepš & Šmilauer, 2003). Scores from the first DCA axis, which accounted for 11.5% of total variation in the dataset were used as a measure of similarity between sites (Total inertia = 5.251, cumulative% variance described by 4 axes = 24.7).

Table 2 Vegetation structure and composition variables calculated from 50 m line transect data for 16 plots in the lowlands of the Greater Cape Floristic Region.

Variable	Description	
Plant cover	Total length (m) per transect covered by vegetation (excludes open ground, with litter, woody debris and sparse seedling cover).	
Vegetation height	Mean height (cm) of vegetation per transect.	
Flower cover	Total length (m) per transect covered by plants in flower at time of survey.	
Open ground	Total length (m) per transect not covered by plant canopy (includes open ground with litter, woody debris and sparse seedling cover)	
Plant composition	Index of similarity between all plots based on plant species composition using first axis scores from a detrended correspondence analysis	
Plant richness	Estimated plant species richness per plot from Incidence Coverage Estimator, ICE (Chao et al., 2000)	
Flower richness	Estimated species richness of flowering plants per plot from Incidence Coverage Estimator, ICE (Chao et al., 2000)	
Annuals	Total length (m) per transect covered by annual species per plot. Plant species classified according to POSA (South African National Biodiversity Institute, 2009)	
Perennials	Total length (m) per transect covered by perennial species per plot. Species classified according to information in POSA.	

The effects of vegetation structure and composition were tested in CANOCO using a canonical RDA of anthophile assemblage data. Unlike the plant data, the segment lengths of an initial DCA indicated a linear distribution, more suited to an RDA (Lepš & Šmilauer, 2003). The ordination was initially constrained by the nine vegetation variables. Stepwise selection was then used to select a subset of the four best fitting variables for the final model, based on partial Monte-Carlo permutation tests to assess the usefulness of each potential variable (Lepš & Šmilauer, 2003). Variance partitioning (Borcard, Legendre & Drapeau, 1992) was used to calculate the relative contributions of the final four variables following procedures in CANOCO described by Lepš & Šmilauer (2003).

Results

A total of 22,352 anthophile individuals were sampled, falling into 198 species. For plants, a total of 348 species were recorded.

Species richness estimates

Observed and estimated plant species richness varied widely between plots (Table 3). Analysis of variance analysis did not detect any difference in estimated plant species richness between plots (F-value = 1.729, p = 0.22). The lowest observed flowering plant species richness in a plot (23 spp.), was recorded at EL2 and the highest (82 spp.) at VG1. Overall, mean (±1 SE) number of observed species in a plot was 42.25 (±4.25). Estimated species richness (ICE) in a plot ranged from 25 spp. (EL2) to 93 spp. (VG1), with a mean ICE of 61.13 (±5.37) in a plot.

Table 3 Non-parametric species-richness estimates using an abundance based species richness estimator, the Incidence Coverage Estimator (ICE) for (a) flowering plants and (b) flower-visiting insects from 16 plots in the lowlands Greater Cape Floristic Region.

(a) Flowering plants	
	N†	Obs.	ICE (±S.D.)	
CO1	3	29	36.31 (±12.94)	
CO2	3	20	32.53 (±12.52)	
EL1	3	57	87.15 (±29.23)	
EL2	3	23	25.11 (±4.87)	
EL3	3	34	73.19 (±27.31)	
EL4	3	39	75.53 (±18.98)	
HE1	3	56	72.69 (±17.28)	
HE2	3	43	63.82 (±18.37)	
HE3	3	42	52.18 (±8.64)	
KL1	3	54	89.67 (±39.63)	
GB1	3	45	53.9 (±14.85)	
GB2	3	62	78.05 (±33.52)	
GB3	3	18	38.3 (±15.17)	
VG1	3	82	92.64 (±18.56)	
VG2	3	40	62.63 (±22.37)	
VG3	3	32	44.34 (±13.03)	
(b) Anthophiles	
	N‡	Obs.	ICE (±S.D.)	
CO1	3	57	70.52 (±15.52)	
CO2	3	31	55.55 (±17.11)	
EL1	3	41	51.08 (±13.22)	
EL2	3	38	47.54 (±10.58)	
EL3	3	35	45.04 (±8.23)	
EL4	3	35	37.84 (±7.75)	
HE1	3	40	56.66 (±13.12)	
HE2	3	60	64.5 (±13.24)	
HE3	3	46	63.17 (±23.15)	
KL1	3	33	45.3 (±13.28)	
GB1	3	27	47.11 (±20.56)	
GB2	3	26	39.41 (±15.93)	
GB3	3	31	39.48 (±12.81)	
VG1	3	19	36 (±16.03)	
VG2	3	40	62.63 (±22.37)	
VG3	3	32	44.34 (±13.03)	
Notes:

Obs = observed number of species.

† Number of vegetation transects used to generate species richness estimates.

‡ Number of pan trap-arrays used to generate species richness estimates.

There was similar variation in recorded and estimated species richness of flower-visiting insects ranging from a minimum of 19 observed species at VG1 and a maximum of 60 observed species at HE2. A mean (±1 SE) of 36.94 (±2.67) species was observed across all plots. Estimated richness (ICE) was highest at CO1 with 71 species, and lowest at VG1 with 36 species (Table 3). Mean ICE across all plots was 38.28 (±2.30). Analysis of variance analysis did not detect any difference in estimated insect species richness between plots (F-value = 1.182, p = 0.38).

Species diversity

There was no significant relationship between insect and plant diversity (t-value = −1.10, p = 0.28).

Species turnover

The RELATE analysis between the plants and anthophiles was significant (ρ = 0.444, p < 0.003), suggesting significant concordance between the assemblages. However, turnover was weaker in plants (ρ = 0.601, p < 0.001) than in anthophiles (ρ = 0.883, p < 0.001).

Power regressions showed that for the entire set of 120 pairwise comparisons, there was a significant positive relationship in turnover for both plant (R2 = 0.443, p < 0.001, Fig. 2A), and anthophile diversity (R2 = 0.709, p < 0.001, Fig. 2B) with increasing plot distance. The plot pairs were spatially separated, but given the relatively small distances (<80 km) from a biogeographical point of view, there were great differences between how the different plot pairs shared plant species (Fig. 2A) and anthophiles (Fig. 2B) at the various distances, with pairwise comparisons separating out into three distinct distance classes, <10 km apart, 20–40 km apart and 65–80 km apart (Figs. 2A and 2B). Furthermore, there were significant positive relationships in turnover within the different distance classes except for the 65–80 km group for plants (Fig. 2A) and 20–40 km group for anthophiles (Fig. 2B), which were not significant. Overall, these results point to great turnover of species even at relatively small geographical distances.

Figure 2 Species turnover of plants (A) and insects (B) as represented by Sorensen pair wise dissimilarity between plot pairs at four spatial scales in the lowlands of the Greater Cape Floristic Region.

Each point represents a pair of sites (120 possible combinations). Dashed line indicates best fit of a power curve for all points, while solid lines are best fit for each subgroup of pairs in three distance classes: 0–10, 20–40, 60–80 km. Number of pairs per distance sub-group are: Blue: N = 40; Yellow: N = 32; and Grey: N = 48. All regression lines are shown (ns = not significant, **p ≤ 0.001, ***p ≤ 0.000).

Effect of plant species composition on insect species composition

Plant species composition did not have a significant effect on anthophile species composition (F-value = 2.92, p = 0.136). The results of RDA revealed that eight plant species explained 60% of the variation in anthophile species composition, geographical distance explained 39% and 1% remained unexplained.

Effect of vegetation structure on insect community composition

Of the nine variables tested, stepwise selection showed that plant species composition, flower cover, plant species richness and average vegetation height were the four most influential variables, collectively explaining 51% of the total variation in the anthophile assemblage data (Fig. 3). Variance partitioning suggested that flower cover was the most important variable, accounting for 22.1% of variation followed by plant composition (10%), mean vegetation height (6.9%) and plant species richness (3.4%). A further 8.7% of the variation could not be attributed to any particular one of these variables.

Figure 3 Biplot from the RDA of anthophile assemblages at 16 sites in the lowlands of the Greater Cape Floristic Region.

Sizes of the circles indicate relative species richness for each site. Arrows indicate the best subset of four vegetation structure variables chosen by forward selection during ordination. For each variable, the relative contribution to the total variation of 51.1% explained by the canonical axes is given in parentheses. Sum of all canonical eigenvalues = 0.511, Monte Carlo permutation test for all axes, F = 2.868, p = 0.006.

Discussion

Species turnover

Dissimilarity between distance plot pairs increased similarly for both insects and plants, with increasing plot pair distance in the subgroups. This is consistent with other findings on other insect functional groups in the GCFR (Procheş & Cowling, 2006; Wright & Samways, 1998; Procheş et al., 2009). There was also high turnover of insect species across the landscape, and to a lesser extent the flora. Even nearby sites showed a high degree of distinctness, with no site sharing more than 29% of plant species and 35% of anthophile species, suggesting high spatial heterogeneity for both groups.

Species turnover was apparent across increasing distance with distinct differences observable from the local (<10 km) to the regional (>60 km) distances. Species turnover, even at a local distance, was high, and was higher in insects than plants. Interestingly, in plants, turnover decreased with greater distance between plot pairs, from local to regional. In contrast, with insects there was high turnover at both the local and regional distances, with a tendency to increase across distance groups. This pattern is most likely a result due to the distances between the distance groups. Firstly, the distance ranges between groups A (Cordoba; Vergelegen, Helderberg, South Africa) and C (Elandskloofberge, 66–73 km) and groups B (Klipfontein and Groenlandberg Cconservancy) and C (68–77 km) overlap. Secondly, the study sites were intentionally placed in sites with near-natural, transformed, remnant or disturbed vegetation. Thus in all distance groups all vegetation statuses are represented.

Caterino (2007) found high levels of spatial variation in beetles across three ecoregions in the California Floristic Province, and concluded that this may be a general characteristic of insect assemblages in Mediterranean-type ecosystems. In Caterino’s (2007) study, plant assemblages and their associated anthophiles, both the local and regional distances had congruent patterns of turnover between sites, but incongruent patterns of species richness. In that study too, there was high spatial variation of anthophiles, and there were no clear patterns in species richness. However, in contrast to Caterino’s (2007) study, our results showed that while the overall pattern of turnover was similar in plants and anthophiles, the patterns within distance classes varied, with the patterns between plants and insects diverging with greater plot-pair distance classes.

Congruence between plant and insect assemblages

We found a strong (44%) positive relationship between plant and anthophile turnover, as has been found in Europe (Ebeling et al., 2008; Papanikolaou et al., 2017), suggesting that such a relationship is geographically widespread. However, in terms of concordance, this means that the areas with similar plant assemblages do not necessarily share similar insect assemblages. This decoupling between the two groups means that plant diversity alone is not a reliable surrogate for insect diversity, at least at the various distance scales examined here. Indeed, plant-insect relationships are highly variable across biomes, scales and insect guilds, suggesting that in each case, different factors may drive insect diversity (Procheş et al., 2009).

Factors affecting diversity of flower-visiting insects

Given the diverse range of taxa encompassed by the entire assemblage of anthophiles, it is difficult to generalise about which factors are most important. The four most influential variables here are likely to represent some of the resource needs of this assemblage, but do not account for all of the observed variation. The fact that flower cover (a measure of the relative abundance of resources for anthophiles) was far more important than plant richness and diversity suggests that resource availability is an important determinant of flower-visitor diversity and abundance, particularly at the local scale (Hegland & Boeke, 2006).

The species rich, temperate flora of southern Africa has a remarkable prevalence of highly specialised pollination systems (Johnson & Steiner, 2003; Pauw & Stanway, 2015), so it may seem strange that plant species composition is not a reliable estimator of anthophile species composition. However, functional relationships between plants and anthophiles are characterised by a high degree of asymmetry (Trøjelsgaard & Olesen, 2013). Anthophiles visiting a specialised plant can be taxonomically diverse, although even specialised pollinators may visit a range of non-specialised plants, they often look for similar amino-acid based resources in plants which confer physiological advantages upon them (Nepi, 2014). The degree of ecological specialisation observed at any one time can be affected by various spatial and temporal factors (Petanidou & Potts, 2006; Fontúrbel et al., 2015), meaning that plant and pollinator species composition may not necessarily be tightly coupled. Although resources for anthophiles are affected by the richness and composition of the local flora, the abundance and quality of suitable resources is not always directly related to plant species richness alone.

Conclusions

To summarize, we compared species richness and turnover relationships between flowering plants and flower-visiting insects across geographic distance (0.5–80 km) in a biodiversity hotspot, the GCFR, South Africa. While we found there to be significant concordance between plants and anthophile assemblages (ρ = 0.444, p < 0.003), turnover was weaker in plants (ρ = 0.601, p < 0.001) than in anthophiles (ρ = 0.883, p < 0.001), and decreased with greater geographical distance between plot pairs. In contrast, insect turnover remained high with increasing geographical distance between plot pairs. Furthermore, flowering plant species composition did not have a significant effect on anthophile species composition (F-value = 2.92, p = 0.136). The discordance between the results here and those of other studies such as Procheş et al. (2009), as well as the inconsistencies noted by those authors, indicate that the factors affecting distributions differ between various taxonomic groups and can confound attempts to draw general conclusions about the relationships between plant and insect species-richness. These findings suggest that while patterns of plant diversity and distribution shape flower-visiting insect assemblages, they are not reliable surrogates.

These results have considerable conservation significance. Firstly, insects must be more densely sampled than flowering-plants to ascertain their full spatial diversity. Secondly, conserving plants in the various parts of this species-rich biodiversity hotspot does not guarantee that all the pollinating insects will also be conserved, with the insects, in effect, being more fragmented than the plants. On the other hand, to conserve all the insects in the area, more land must be set aside for them, while on the other, certain specialist plants may not have their pollinators present. Overall, specialist plant-insect mutualisms are more vulnerable than generalist ones. Conservation activities that improve functional connectivity across the overall landscape will help maintain these plant-insect mutualisms. This may include establishing conservancies where set-aside land is an intrinsic part of the agricultural landscape. Intercropping using a range of vegetation is extensively employed but a move to vegetate towards indigenous fynbos could be instigated to a greater extent. These approaches are feasible at least in vineyards (the dominant agricultural type in this area) under the Biodiversity and Wine Initiative (http://wine.co.za) with which many vineyards have partnered.

Supplemental Information

Supplemental Information 1 Appendix S1. Distance table for all plots used in species turnover study.

Click here for additional data file.

Supplemental Information 2 Appendix S2. Insect and plant species identified from the study.

Click here for additional data file.

Supplemental Information 3 Pan and plot assemblage similarity.

Description of the results of the pan and plot similarities.

Click here for additional data file.

Supplemental Information 4 Pan and plot assemblage similarity.

Classification of sixteen sites in the lowlands of the Cape Floristic Region. Classification is based on the similarity of flowering plant assemblages recorded from vegetation transects (a), and the similarity of flower-visiting insect assemblages recorded from coloured pan trap surveys (b). Significant (p ≤ 0.05) branches or groups of plots, as tested using SIMPROF analysis, are indicated by the black horizontal lines.

Click here for additional data file.

We thank C. Eardley, S. van Noort and H. Geertsema for taxonomic advice. We would also like to thank the two anonymous reviewers who have taken the time to read and give critical input on the manuscript.

Additional Information and Declarations

Competing Interests

Author Contributions

Field Study Permissions

Data Availability

The authors declare that they have no competing interests.

John P. Simaika analysed the data, contributed reagents/materials/analysis tools, prepared figures and/or tables, authored or reviewed drafts of the paper, approved the final draft.

Michael Samways conceived and designed the experiments, contributed reagents/materials/analysis tools, authored or reviewed drafts of the paper, approved the final draft.

Sven M. Vrdoljak performed the experiments, analysed the data, prepared figures and/or tables, authored or reviewed drafts of the paper, approved the final draft.

The following information was supplied relating to field study approvals (i.e. approving body and any reference numbers):

Permissions for conducting the study were obtained from the relevant authority at: Helderberg Municipal Nature Reserve and Hottentots Holland Provincial Nature Reserve—permission from Cape Nature (No. 372/2003).

The following information was supplied regarding data availability:

Simaika, John (2018): Plant and insect data. figshare. Fileset. https://doi.org/10.6084/m9.figshare.6236759.v1.

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
