# Peer review of "Species turnover in plants does not predict turnover in flower-visiting insects"

_PeerJ, doi:10.7717/peerj.6139_

## Round 0.1 · original submission · Major Revisions

Please, when revising your paper take into account comments made by the reviewers. In particular rev#2 requires a revision of the method section and rev#1 remarks that conclusions should focus more on the results of the manuscript.

Reviewer 1 ·

Basic reporting

Simaika et al. investigated whether flowering plant diversity can be used as a surrogate for the diversity of flower visiting insects in the Cape Floristic Region, South Africa. They found that plant diversity was a poor predictor for anthophile diversity although there was similarly strong spatial turnover in species assemblages.

The manuscript is in all parts well-written and well-structured. The literature cited is adequate and the results are relevant to the research questions. I just have few minor suggestions to improve the structure of the manuscript (for comments on the structure of the discussion see “Validity of findings”):

L43-59 I think this paragraph could be shortened, because it focuses on plants and insects in general while the paper is focused on flower visiting insects.

L91-93 It could be better explained why the spatial scale is important for the research question and conservation planning.

Experimental design

The research questions are interesting and the experimental design is valid and meaningful. Additionally, the methods are generally well described and provide important details. However, some questions remain about the sampling and methods:

L147-154 Here you should describe how you measured flower cover. Additionally, you sampled the plots twice during a three month period, but flowering status was only recorded once, before the insect sampling? Could the flower cover have changed dramatically from the vegetation to the insect sampling? Is it also possible that other herbaceous plant species appeared later in the season that were not recorded during the plant sampling?

L.168-171 Please, provide more details. How many models were conducted? Which were the response variables? Did you account for the nested design of your data?

Validity of the findings

I have no experience with some of the statistical methods used such as cluster analysis, seriation and RDA, but the authors’ explanations are convincing to me.

However, for the simple linear regression testing the relationship between plant and insect diversity the nested design of the study (with several plots per site) should be taken into account, because several plots from the same site are not independent. Mixed models would be appropriate in this case.

The discussion addresses all important issues, but I would prefer to structure the discussion according to the two research questions stated at the end of the introduction. Additionally, I would be more interested in how the spatial scale is important. The title states that spatial scale matters when flowering plants are used as surrogates for anthophiles suggesting that plants can be used as surrogates at certain scales, but not at others. However, this question is not addressed in the manuscript. You should change the title or focus more on the importance of the spatial scale for the usage of plants as surrogates for anthophiles.

Conclusions: The conclusions should focus more on the results of this manuscript. You should repeat the main findings of your work and what we learned from this study. Additionally, you could also mention how your results affect the conservation planning in your region. Is the current conservation management good? What could be improved to promote flower-visiting insects? Should new conservation schemes be developed that target insects directly?

Additional comments

Other minor comments:
L.23 This sentence suggests to me that the diversity of plants was related to the diversity of anthophiles which was not the case. Consider rewording.
L60-61 Should not plant diversity predict insect diversity?
L83-84 The cited paper is only on herbivores. So maybe better write “established plant-insect herbivore diversity relationship”?
L120 Here you could already mention Appenidx S4 (In the document peerj-28146-Appendix_S4_Distance_table_for_all_plots.docx the title is Appendix S1?).
L.162-163 Species richness of plants and anthophiles? I think you don’t need to repeat that insect data was pooled across transects per plot.
L.218 This table should be mentioned in the methods section.
L.241 “Clear pattern” regarding which question?
L.253 Do you mean “plots”?
L.267-270 Why are the results of Fig. 2b not in the same paragraph as Fig. 2a?
L.313 According to which statistical test?
Fig.2 I think you don’t need to repeat the letters in parentheses: (a) and (A).
Appendix S1: It would be interesting to sum the abundances of each insect species to see which taxa were dominant.

Reviewer 2 ·

Basic reporting

no comment

Experimental design

no comment

Validity of the findings

no comment

Additional comments

The Authors compared insect diversity and abundance with plant diversity at six locations (16 plots) in protected areas of Cape Floristic Region, South Africa. The purpose of the analysis was evaluating the reliability of plant diversity as a surrogate for insect diversity/abundance. This has relevance for conservation (both in the region and in general), since plant diversity is often better known than insect diversity. The Authors claimed to have performed investigations at different spatial scales (from 0.5 to 80 km). Their main findings are that: 1) plant diversity is not correlated with insect diversity and abundance, and hence does not constitute a useful proxy to be used in conservation study; and 2) plant and insect turnover is similar, and strongly affected by ‘spatial scale’, in terms of pairwise distance between localities. I have a few minor comments I have annotated while reading the paper, and a major concern about the Authors’ conclusions.
Looking at the map of sampling locations, it is apparent that the Authors focused in fact on three distinct areas (A1 = circles + triangles + diamonds; A2 = crosses + stars; A3 = squares). It is true that the total area of the analysis is not extremely large, but it is also true that the region is particularly rich in species, and that differences are expected to emerge at relatively small spatial scales. The distribution of sampling sites makes the results in Fig. 3 (which seems to me as the core result of the paper, also motivating the title) not very surprising; in fact, the graph shows simply that the turnover within area groups (A1, A2, A3) is lower than the turnover between A1 and A2 (which are at approximately at 20-30km of distance), and that the highest turnover is recorded between A1+A2 vs. A3 (which are at a distance of 70-80km). Now, that is far from being surprising, being simply an additional confirmation to the commonness of distance-similarity decay relationships. The fact that this applies to both insects and plants has no clear, straightforward implications for conservation.
Thus, in general, although I think that the dataset collected by the Authors is valuable, and the paper is relatively clear and well written, I am not convinced it offers some novel insights regarding turnover patterns.
Yet, although a negative result, the lack of correlation between insect diversity and plant diversity is a valuable result (even if the number of plots, considering also their geographical clustering, is a bit small to draw general conclusions). My advice would be that of reworking the paper by focusing on this aspect, also putting emphasis on all possible caveats.
Also, I found the method section not particularly clear. The Authors provide many technical details on the statistical procedures, including the software tools used to perform the analyses, which is appreciable. However, the reasons why each analysis was chosen is, in most cases, not provided. Authors should better explained the goals behind their choices, i.e. clarify a priori the questions addressed by each analysis. Considering my general advice above, I would recommend to simplify the analytical approach; many of the analyses presented in the text seem a bit out of the main goals described in the introduction.








66 specify the level of endemism
144-146 needs clarification
167 remove comma before the bracket
167-176 How seriation is computed could be better explained
178-179 What is the point of computing shared species between plot pairs? This quantity is not independent from plots’ species richness (which is affected by factors likely independent from overlap itself), and thus does not provide an informative measure of overlap. I would have expected to see the quantity compared with some null expectation obtained by randomly reallocating species in plots.


228 In general, I have no problem in the use of simple linear regression as a robust approach to detect relationships; however, in the case of the comparison between insect and plant richness, it would make sense to check at least for rank correlation, especially considering that the dependence between plants and insects, and the likely asymmetric specialization between the two groups could possibly lead to different trends in species richness across sites.
Also, I think that readers (at least myself) would like to see a figure where insect diversity and abundance is compared with plant diversity across sites.

Scale of x-axis in fig 3 is missing (is it km x 10-4?)
Discussion, lines 287-289: “Interestingly, in plants, turnover decreased with greater distance between plot pairs, from local to regional. In contrast, with insects there was high turnover at both the local and regional scale, with a tendency to increase across distance groups.”
This is not apparent from fig. 3, and a bit in contrast with the positive relationship claimed elsewhere in the manuscript.

---

## Round 0.2 · Major Revisions

The reviewer and I concur on the fact that many aspects of the manuscript need further revision. Please, when preparing your revised version of the manuscript, carefully consider the changes suggested.

Reviewer 1 ·

Basic reporting

The manuscript is well-structured and well written. However, I see a major problem with the hypotheses. Both hypotheses include the question how plant diversity affects insect diversity. However, there is no statistical test on diversity patterns, but only on community composition.

Experimental design

The experimental design is clear, relevant and meaningful.

Validity of the findings

I do not think that the analyses support all conclusions. See "Basic reporting" and "comments for the author" for details.

Additional comments

I appreciate the effort of Simaika and colleagues to revise the manuscript. Yet, many aspects of the manuscript are still unclear and need further revision.

First of all, throughout the manuscript the usage of the terms species richness (or diversity) and community composition are not clearly separated. In several parts of the manuscript the authors draw conclusion about diversity patterns based on analyses of community composition (see comments below). It remains also unclear why it is important to compare species turnover of plants and insects as the introduction is mainly about diversity patterns and “turnover” is mentioned in the hypotheses for the first time.
In the discussion the authors write that based on their study the diversity of plants cannot predict the diversity of insects (L285-286). However, there is no statistical test showing whether plant diversity is a good predictor for insect diversity (again community composition changes are not the same as diversity changes). I agree with Reviewer 2 that the comparison of plant and insect diversity is the most interesting part of the study and therefore a statistical test is urgently needed to support this result. Based on the sampling design with different distances between the sampling sites the authors should account for this spatial dependency, e.g. by adding random effects or by including coordinates of the sites in the model.

Comments:

L48 I do not understand this sentence. Why is stability needed when plants and insects react similarly to climatic conditions as written in the previous sentence? Generally, I think that the impact of past climate is off topic and should be rather excluded. Instead the focus should be more on the concept of species turnover and why it is important for conserving affectively both, plant and insect species.

L88-89 In the analysis the authors do not compare the diversity at different distance classes, but the dissimilarity of community compositions. Diversity and community composition are totally different concepts.

L152 “data pooled per array and visit”?

L154-155 I assume you used the same estimator for plants and insects?

L174-177 After reading the cited reference I understand that 1 means that the samples that are closest to each other have the most similar communities while the most distant sites would have the least similar communities. I think this should be explained in the manuscript. Additionally, from the results I understand that you also compared plants and insects in this analysis. You should explain how this was done.

L184 This header is misleading as you do not relate plant and insect diversity, but their community compositions.

L219-220 “Trend” according to which hypothesis?

L228-229 There is no test showing the relationship between plant and insect species richness.

L231-242 I find this analysis not very interesting. It is not very surprising that the closest communities are generally more similar. I would omit this analysis or move it to the supplementary material.

L266 replace by “Effect of vegetation structure on insect community composition”

L285 See comment above, there was no test for species richness relationship.

L286-291: Unfortunately, the patterns found in Fig. 3 are not explained in the discussion. What could be the reasons for these patterns? I agree with Reviewer 2 that Figure 3 represents the 3 main spatial groups of sites as shown in Fig. 1 (A: rectangles, B: circles, triangles and diamonds, and C: star and crosses). The points within the first distance class are comparisons of sites within each of the 3 groups (A,B and C). The distance class of about 25-40km represents comparisons of sites of group B and C. In the last distance group (60-70km) the authors compare sites of group A with both B and C. Here the authors interpret the results that turnover of insects tends to increase across distance groups. However, the authors might have found these results simply because B is a little bit closer to A than C and it could be another environmental factor that influences the higher similarity of B to A than C to A (for example percentage agricultural land in the surrounding or crop grown in the vicinity). This possibility should be discussed.

L302-313 The authors should also acknowledge that there are many studies showing that plant species richness is very important in predicting pollinator species richness (e.g. Ebeling et al., 2008. How does plant richness affect pollinator richness and temporal stability of flower visits? Oikos 117, 1808–1815. or Papanikolauo et al 2017. Wild bee and floral diversity co-vary in response to the direct and indirect impacts of land use. Ecosphere 8(11):e02008)

L303-304 I do not understand this sentence.

L306 Specify what was analyzed: “herbivore community composition”?

L327 Again, there is no test to support any conclusions related to diversity patterns.

L344-346 This test also includes community composition and not diversity.

L358-361 I do not fully understand the last sentence, please simplify.

Conclusions: You found that flower cover had the highest impact on insect communities. Are there any management practices that can increase flower cover?

---

## Round 0.3 · Minor Revisions

I find that the revised version of the manuscript has been greatly improved and it fulfilled most of the suggestions of the reviewers. I still have a minor remark, regarding the relevance of using turnover.
I agree with the reviewer that: "It remains also unclear why it is important to compare species turnover of plants and insects as the introduction is mainly about diversity patterns and “turnover” is mentioned in the hypotheses for the first time."

In my opinion, a more detailed introduction on the relevance of turnover in the context of this study may give additional value to the paper.

---

## Round 0.4 · accepted · Accept

Many thanks for this further revision of your manuscript. i think that the relevance of beta diversity in relation to the core idea of your paper (i.e. "the interrelationship between two mutually dependent groups, and how this relationship is affected by geographic distance") is now very clearly stated.

#